# Automatic Loss Function Search for Predict-Then-Optimize Problems with Strong Ranking Property

**Boshi Wang**[1][*] **Jialin Yi**[2][*] **Hang Dong**[3] **Bo Qiao**[3]**, Chuan Luo**[3][†] **Qingwei Lin**[3][†]

[1]The Ohio State University, United States
[2]London School of Economics and Political Science, United Kingdom
[3]Microsoft Research, China
`wang.13930@buckeyemail.osu.edu, j.yi8@lse.ac.uk,`
`{hangdong, boqiao, chuan.luo, qlin}@microsoft.com`

## Abstract

Combinatorial optimization problems with parameters to be predicted from side information are commonly seen in a variety of problems during the paradigm shifts from reactive decision making to proactive decision making. Due to the misalignment between the continuous prediction results and the discrete decisions in optimization problems, it is hard to achieve a satisfactory prediction result with the ordinary $l_2$ loss in the prediction phase. To properly connect the prediction loss with the optimization goal, in this paper we propose a *total group preorder* (TGP) loss and its differential version called *approximate total group preorder* (ATGP) loss for predict-then-optimize (PTO) problems with strong ranking property. These new losses are provably more robust than the usual $l_2$ loss in a linear regression setting and have great potential to extend to other settings. We also propose an automatic searching algorithm that adapts the ATGP loss to PTO problems with different combinatorial structures. Extensive experiments on the ranking problem, the knapsack problem, and the shortest path problem have demonstrated that our proposed method can achieve a significantly better performance compared to the other methods designed for PTO problems.

## 1 Introduction

Many decision making processes under uncertainty in real world are solving combinatorial problems with parameters unknown to the decision maker. A traditional method to address the uncertainty issue is to add assumptions on the distributions of parameters (Hentenryck & Bent, 2006). Alternatively, a recent approach is to predict the unknown parameters from the correlated features using machine learning methods and solve the combinatorial problems based on the predictions. This paradigm, which is called *predict-then-optimize* (PTO), has been widely employed in practice systematically (Elmachtoub & Grigas, 2021). For example, Google Maps estimates the travel times of roads in the traffic to compute a shortest path (Lau, 2020); Computer clusters predict the processing times and resource demands of computation tasks to make good job scheduling on servers (Mao et al., 2016); Hedge funds forecast the return rates of different stocks to optimize their portfolio in the next trading day (Thomson, 2021).

However, the commonly used $l_2$ loss function (*i.e.,* $l_2$ distance between predictions and true values) usually cannot achieve ideal decision results in predict-then-optimize problems (Demirović et al., 2019). This misalignment between the $l_2$ loss in prediction and quality of the decision comes from the misalignment between the continuity on the prediction problem and the discontinuity on the structure of the combinatorial optimization problem.

A straightforward remedy of the above misalignment problem is to adjust the loss to reflect the difference in objective values of the optimization solutions generated using the predicted and observed

---

[*]Equal contribution. Work done during internship in Microsoft Research Asia.
[†]Corresponding authors.

Figure 1: Mechanism of our APOC algorithm.

parameters, called *Smart "Predict, then Optimize"* (SPO) loss (Elmachtoub & Grigas, 2021). However, due to the combinatorial nature of the optimization problems, the SPO loss is usually piecewise flat and has multiple discontinued points. As a result, the derivatives of SPO loss is either zero or nonexistent, which prohibits the training of gradient-based deep learning algorithms. Besides, most solvers for combinatorial optimization problems are not differentiable, which cannot be directly incorporated with the widely adopted gradient-based learning approaches nowadays.

Current approaches to minimizing the SPO loss can be categorized into two classes. The first one is to differentiate the discrete optimization solver by approximating the objective with a certain family of functions, and this series of methods works when the combinatorial optimization problem is linear (Wilder et al., 2019a;b; Pogančić et al., 2019). The other class of research tries to propose new surrogate loss functions for SPO loss, *e.g., SPO+* loss for linear programming (Elmachtoub & Grigas, 2021) and piecewise linear loss for dynamic programming (Stuckey et al., 2020). However, these surrogate loss functions are problem-specific (*i.e.,* depending on the type of the combinatorial optimization problem) and require much domain knowledge from experts to design.

Instead of designing the surrogate loss metric for each optimization problem manually, an alternative approach is to find a good loss metric from a predefined search space in an automatic manner, which is inspired by the recent progress in automated machine learning (AutoML) (Zoph & Le, 2017; Pham et al., 2018; Liu et al., 2018). These AutoML approaches usually define an appropriate search space and then conduct some search algorithms, *e.g.,* reinforcement learning and genetic algorithm, to find a good metric. Li et al. (2019) and Li et al. (2021) studied the automatic loss function search problem for computer vision tasks. They defined the search space by replacing the mainstream metrics for semantic segmentation with their differentiable approximations. However for predict-then-optimize problems, there is no such evaluation metrics available except for SPO loss. Moreover, these metrics, which are designed for segmentation tasks in computer vision domain, are not suitable to evaluate the prediction results for the combinatorial optimization problems.

In this paper, we propose a framework of automatic loss function search called APOC (Automatic Prediction and Optimization Connector) to tackle a wide range of predict-then-optimize problems whose optimal solution is determined by the total group preorder of the parameters of the combinatorial problem, called strong ranking property (see for Definition 2 for details.). These problems have the same optimal solution for different sets of parameters, as long as those sets of parameters preserve the same total group preorders. The key idea is to build the proper search space in which different loss functions capture the partial comparisons between different groups of the parameters of the optimization problem.

Our contributions are summarized as follows: 1) We theoretically prove that $l_2$ loss is not an ideal choice in a linear regression setting of PTO problems; 2) We propose the total group preorder loss for PTO problems and relax it to a differentiable approximated version, which is fitted using a family of *Derivative-of-Guassian* wavelet functions to capture the group comparison relationship among the items with variable sizes and weights; 3) We propose an effective method to automatically search for a differentiable surrogate loss based on approximated group total preorders of items for PTO problems called APOC; 4) The proposed APOC method has been validated on three classic combinatorial problems with strong ranking property.

## 2 RELATED WORK

**Predict-then-optimize (PTO) problems** capture a common pipeline of machine-learning-assisted optimization solvers: *predicting* the unknown parameters of the optimization problem from the con-

textual information and then *optimizing* based on the predicted parameters (Elmachtoub & Grigas, 2021). In practice, PTO problems have a broad range of real-world applications Lau (2020); Mao et al. (2016); Thomson (2021); Luo et al. (2020; 2021). However, many combinatorial optimization problems have piecewise flat value function landscapes which prohibit an end-to-end learning with gradient-based prediction models.

One line of work is to differentiate the optimization solvers for specific problems and embed them as layers into the network architecture through which the gradients can be propagated, which originates from the differentiable optimization (Amos, 2019). Grover et al. (2018) proposes a sorting network for ranking problems by relaxing the permutation matrix output by sorting algorithms to the unimodal row-stochastic matrix. For the node clustering problem, Wilder et al. (2019b) differentiates the K-means algorithm by assigning nodes to clusters according to a soft-min function. Pogančić et al. (2019) interpolates the value of function of linear programming using piecewise linear functions and embeds as layers the solvers of linear programming.

The other line of work focuses on searching for a surrogate loss function to approximate the value function of the optimization problem, so-called *regret* in decision theory (Bell, 1982). Elmachtoub & Grigas (2021) derives a convex upper bound on the regret of linear programming, called SPO+ loss. Wilder et al. (2019a) proposes the QTPL loss by adding a quadratic penalty term to the continuous relaxation of the regret such that results from differentiating over quadratic programs can be used. Mandi & Guns (2020) overcomes the non-differentiablity of the regret by adding a logarithmic barrier term, called IntOpt. Stuckey et al. (2020) approximates the value function of dynamic programming problems as piecewise linear functions with learn-able parameters. Yoon et al. (2013) proposes *mean objective cost of uncertainty* (MOCU), the difference between the value functions of a robust solution and the optimal solution, to evaluate the uncertainty of parameters in decision making, *e.g.,* experiment design (Boluki et al., 2018) and active learning (Zhao et al., 2021).

**Loss function** guides the machine learning algorithms to produce good predictions for different tasks, which is usually designed with domain knowledge from experts (Masnadi-Shirazi & Vasconcelos, 2008; Bruch et al., 2019). Automatic search of suitable loss function without domain knowledge has recently received much attention from the computer vision community. Li et al. (2019) uses reinforcement learning algorithms to learn better loss functions with good generalization and transfer-ability on different vision tasks. Wang et al. (2020) adopts both random search and reinforcement learning algorithms to search for loss functions on face recognition problems. Li et al. (2021) explores the possibility of searching loss function automatically from scratch for generic tasks, *e.g.,* semantic segmentation, object detection, and pose estimation. However, the search spaces in these methods are specially designed for vision tasks and cannot be directly applied to PTO problems. Most work on automatic loss search follows the searching algorithms used in AutoML. We refer to He et al. (2021) for comprehensive survey on searching methods in AutoML. Natural questions are: 1) whether we can design a suitable search space for loss functions in PTO problems; and 2) whether there is a strategy to search adequate loss functions systematically for different optimization problems using techniques from AutoML.

## 3 ATGP LOSS FOR PREDICT-THEN-OPTIMIZE PROBLEMS

### 3.1 MISALIGNMENT IN PREDICT-THEN-OPTIMIZE PROBLEM

The problems we consider are the type of *predict-then-optimize* (PTO) problems in the following formulation. For an optimization problem in the following form

$$\underset{z}{\text{maximize}}\ U_c(z) \qquad \text{subject to } z \in Z \qquad (1)$$

where $z$ is the decision variable, $c \in \mathbb{R}^d$ is the parameter of the objective function $U$, and $Z$ is the feasible set which does not depend on $c$. A decision maker needs to solve (1) without knowing the exact value of $c$. Instead, the decision maker will observe a feature vector $x \in \mathbb{R}^k$. The goal of the decision maker is to *predict* the value of $c$ and then *optimize* (1) based on the prediction $\hat{c}$. To distinguish with the parameters of the prediction models (*e.g.,* weights of neural networks), we call $c$ the PTO parameter.

Instead of focusing on the prediction error between $c$ and $\hat{c}$ (*e.g.,* $l_2$ distance), the decision maker cares more about a high-quality decision result generated from the prediction. Elmachtoub & Grigas

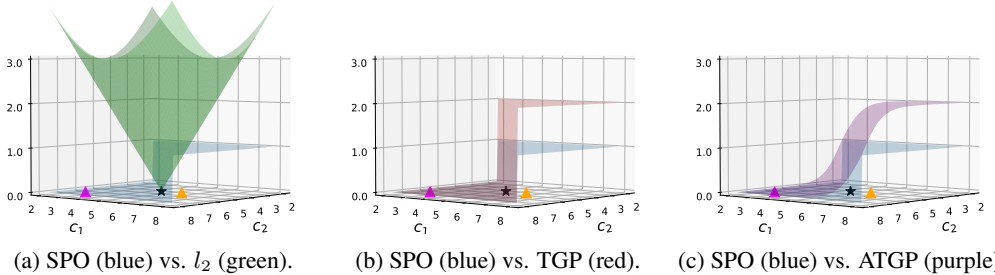

(a) SPO (blue) vs. $l_2$ (green).  (b) SPO (blue) vs. TGP (red).  (c) SPO (blue) vs. ATGP (purple).

Figure 2: Comparing the SPO loss with different surrogate losses. The true item values to be ranked are $(4.9, 5.1)$ (dark star). Two predictions are $(3, 7)$ (magenta triangle) and $(6, 5)$ (orange triangle).

(2021) characterizes this quantity by the *Smart "Predict, then Optimize"* (SPO) loss, which equals to the loss incurred by solving the optimization problem (1) using $\hat{c}$ instead of the true parameter $c$.

$$\ell_{\text{SPO}}(\hat{c}, c) = \mathbb{E}\left[U_c^* - U_c(z^*(\hat{c}))\right]$$

where $z^*(c) = \arg\max_{z \in Z} U_c(z)$ and $U_c^* = U_c(z^*(c))$.

Figure 2a shows the misalignment between the $l_2$ distance (green) and the SPO loss (blue) in a ranking PTO problem. In this ranking PTO problem, there are two items to be ranked. The true values of the items are $c_1 = 4.9$ and $c_2 = 5.1$, which are also the values to predict. The prediction denoted by the magenta triangle has a lower SPO loss than the orange one, and therefore yields a better solution. However, prediction model that minimizes the $l_2$ distance will prefer the orange one (whose $l_2$ distance is lower) and produce a worse solution with regard to the ranking task.

However, the SPO loss of most PTO problems is a piecewise linear function due to the combinatorial structure in the objective function $U_c(z)$ and the solution map $z^*(c)$. As a result, its derivatives in the predicted PTO parameter $\hat{c}$ is either zero or nonexistent, which prohibits the usage of prediction models whose training is based on gradient descent algorithms.

In this paper, we consider a collection of combinatorial optimization problems with *strong ranking property*. For this collection of optimization problems, we propose the *total group preorder* (TGP) loss and its differentiable variant *approximated total group preorder* (ATGP) loss as the training targets in the prediction stage. The ATGP loss will approximate the SPO loss with smooth shape, which is friendly and effective for the gradient-based training manner, as is shown in Figure 2c.

## 3.2 STRONG RANKING PROPERTY AND (A)TGP LOSS

The strong ranking property is a stronger version of the ranking property proposed by Demirović et al. (2019). The ranking property from Demirović et al. (2019) requires that two instances of a combinatorial optimization problem have the same solution if the *pairwise* orders of the parameters of two instances are the same, which captures the importance of total preorder to the optimality of the ranking problems. However, the total preorder is not sufficient to find an optimal solution for many combinatorial problems with *group-wise* comparisons in nature, *e.g.*, knapsack problem with fixed item weights[1] and shortest path problem. To extend this ranking property to these problems, we propose a stronger version of ranking property which takes the group-wise preorder into consideration.

**Definition 1** (Total group preorder). *For a combinatorial optimization problem with parameter $c = (c_1, \ldots, c_d)$, the total group preorder* $\text{TGP}(c) = \Big\{ (A, B) :$ *non-empty subsets* $A, B \subset \{1, \ldots, d\}$ *and* $A \cap B = \emptyset$ *such that* $\sum_{i \in A} c_i \geq \sum_{j \in B} c_j \Big\}$.

For a ranking problems of 3 items with true weights $c_1 = 1, c_2 = 2, c_3 = 3$, then the total group preorder is $\text{TGP}(c) = \{(\{3\}, \{1\}), (\{3\}, \{2\}), (\{2\}, \{1\}), (\{3\}, \{1, 2\})\}$. For many combinatorial

---

[1]The weights of items are fixed constants, as well as the capacity. The values of items are PTO parameters to be estimated. Throughout this paper, we focus on this subset of knapsack problems.

optimization problems, it suffices to know the total group preorder of the parameters to yield the optimal solution.

**Definition 2** (Strong ranking property). *A combinatorial optimization problem satisfies strong ranking property if for any two instances $P$ and $P'$ with parameters $c$ and $c'$ respectively, $\mathrm{TGP}(c) = \mathrm{TGP}(c') \implies P$ and $P'$ have the same solution.*

The strong ranking property indicates that the prediction algorithm only needs to learn the preorders between different sets of groups in PTO parameters. Inspired by the strong ranking property, we propose the *TGP loss* function which captures the difference between the TGP under predicted PTO parameter $\hat{c}$ and that under the true parameter $c$.

**Definition 3** (TGP loss).

$$\ell_{TGP}(\hat{c}, c \mid L) = \|\mathrm{sign}(L\hat{c}) - \mathrm{sign}(Lc)\|_2^2 \tag{2}$$

*where $\mathrm{sign}(x)$ is a point-wise sign function and $L$ is a matrix whose elements can be any real numbers, called TGP matrix.*

To capture the complete TGP, we need $L$ to be $\binom{2^d}{2} \times d$ which needs exponentially growing memory as the number of PTO parameter $d$ increases. However, this can be avoided by noticing that the optimal solutions of different optimization problems are determined by different *subset* of its TGP, *e.g.,* $\{(\{3\}, \{1\}), (\{3\}, \{2\}), (\{2\}, \{1\})\}$ for the ranking problem mentioned above. Instead, we fix the number of rows in $L$ and let each row of $L$ act as a linear filter that extracts out a group-wise comparison within $c$ so that each $L$ represents a subset of TGP; as a result, different optimization problems will have different TGP matrices.

Note that TGP loss is not differentiable because the sign function on $\hat{c}$ is not differentiable, we derive a differentiable version of it by replacing the sign function with hyperbolic tangent function, called *approximated total group preorder* (ATGP) loss.

**Definition 4** (ATGP loss).

$$\ell_{ATGP}(\hat{c}, c \mid L, \gamma) = \|\tanh(\gamma L\hat{c}) - \mathrm{sign}(Lc)\|_2^2 \tag{3}$$

*where $\gamma$ is a temperature hyperparameter that controls the degree that the TGP loss is approximated to and $\tanh$ is a point-wise hyperbolic tangent function.*

**Lemma 1.** *For any $\hat{c}, c \in \mathbb{R}^d$ and TGP matrix $L$, we have*

$$\lim_{\gamma \to +\infty} \ell_{ATGP}(\hat{c}, c \mid L, \gamma) = \ell_{TGP}(\hat{c}, c \mid L)$$

Here the hyperbolic tangent function is used to approximate the sign function with a differentiable functional form. Figure 2c visualizes the ATGP loss for the ranking PTO problem with TGP matrix $L = [1, -1]$ and $\gamma = 0.9$.

## 3.3 ROBUSTNESS OF ATGP LOSS

Despite of the difficulty on theoretically proving the non-optimum of the commonly used $l_2$ loss in the prediction stage, we can prove in a simple setting with linear regression as the prediction algorithm that the ATGP loss is generally better than the traditional $l_2$ loss for the ranking problem. Specifically, ATGP loss is more robust to noise added to the observation of the PTO parameters in the training data set.

**Learning to rank with linear regression** Suppose there is a set of training examples $\{(x^{(i)}, c^{(i)}) : i \in S\}$ in the training set $S$ identically and independently sampled from some distribution $D$. Each example has a feature vector $x^{(i)} \in \mathbb{R}^k$ and a vector $c^{(i)} \in \mathbb{R}^d$ of response variables. Here the response variables are the values of $d$ items that the decision maker aims to rank. The decision maker predicts the value of items using a linear regression model, *i.e.,* compute a matrix $H \in \mathbb{R}^{d \times k}$ such that for a new pair $(x, c) \sim D$, the linear estimator $Hx$ predicts a score for each response variable in $c$ given $x$. The decision maker will rank the items according to the predicted values $Hx$. An example is routing recommendation in which the feature vector $x$ encodes

the context information like weather, time, temperature and $c \in \mathbb{R}^d$ is the congestion levels of the $d$ routes within a map. The goal is to find the $n$ routes with the least congestion levels.

Let the ground truth linear transformation from the feature points to response variables be $H^*$, and the ground truth response variables be $\bar{c}^{(i)} = H^* x^{(i)}$. The noisy response variables that we actually observe are $c^{(i)} = \bar{c}^{(i)} + \varepsilon^{(i)}$ where $\varepsilon^{(i)}$ is the noise i.i.d. over each sample $i$. For a loss function $\ell$, we denote its empirical risk minimizing estimator $\hat{H}_\ell = \arg\min_H \sum_{i \in S} \ell(Hx^{(i)}, c^{(i)})$, so the predicted values given feature $x^{(i)}$ would be $\hat{c}_\ell^{(i)} = \hat{H}_\ell x$. We define the robustness of a loss function as the minimum level of noise such that there exists at least one sample where the predicted values have different total group preorder as the ground truth values, which intuitively stands for the minimum efforts to "corrupt" the system.

**Definition 5** (Robustness). *The robustness $\mathcal{R}(\ell)$ of a loss function $\ell$ is defined to be*

$$\mathcal{R}(\ell) = \inf_{\varepsilon}\{\|\varepsilon\|_2^2 : \exists i \in S \ s.t. \ \mathrm{TGP}(\hat{c}_\ell^{(i)}) \neq \mathrm{TGP}(\bar{c}^{(i)})\} \tag{4}$$

We show that as long as the sample that's most easily corrupted does not share the same low-dimensional subspace of other samples, the robustness of ATGP loss function is no lower than the $l_2$ loss.

**Theorem 1.** *Let $i^*$ be the index of the ground truth optimization parameter which is most sensitive to noise perturbation, i.e., $\inf_\varepsilon\{\|\varepsilon\|_2^2 : \mathrm{TGP}(\bar{c}^{(i^*)} + \varepsilon) \neq \mathrm{TGP}(\bar{c}^{(i^*)})\} \leq \inf_\varepsilon\{\|\varepsilon\|_2^2 : \mathrm{TGP}(\bar{c}^{(j)} + \varepsilon) \neq \mathrm{TGP}(\bar{c}^{(j)})\}, \forall j \in S$. If the linear space expanded by all feature vectors except $i^*$ has low rank structure (i.e., $\mathrm{rank}\left(\{x^{(i)} : i \in S/\{i^*\}\}\right) < k$) and $x^{(i^*)}$ doesn't belong to this space, then there exists a TGP matrix $L$ such that*

$$\mathcal{R}\left(\ell_{ATGP}\right) \geq \mathcal{R}\left(\ell_{l_2}\right)$$

The choice of TGP matrix $L$ depends on the specific type of the optimization problem $U$. A TGP matrix suitable for top-$n$ ranking problem may perform badly on the knapsack problem and shortest path problem. To solve this, we propose a searching algorithm to automatically find the suitable TGP matrices for different optimization problems with strong ranking property.

## 4 Adaptive ATGP Loss Function Search with APOC

To search for the proper ATGP loss functions for different optimization problems, we propose a search algorithm called APOC (Automatic Prediction and Optimization Connector), following the spirit of AutoML (Zoph & Le, 2017; Pham et al., 2018; Li et al., 2021). To make the size of the searching space for the TGP matrix stable and preserve the flexibility of comparing items, we parameterize the rows of TGP matrix with a set of discretized *Derivative-of-Gaussian* (DoG) filters, which are wavelet filters widely used for edge detection in computer vision (Kennedy & Basu, 1999; Zhang et al., 2020). The wavelet filters can well capture the weighted group-wise comparison among the items with variable group size and are updated based on the prediction quality induced by them.

### 4.1 Search Space and Wavelet Filter

Recall that for the PTO parameter $c$, we want $\tanh(\gamma Lc) \approx \mathrm{sign}(Lc)$ to approximately capture different group-wise comparisons between the entries of $c$. To this end, we parameterize the TGP matrix $L$ by a set of sliding discretized *Derivatives-of-Gaussian* (DoG) filters. Each filter has a reception field which will select a subset of entries of $c$, and assigns weights to the selected entries in the reception field, as is shown in Figure 3. The inner product between the selected entries in the reception field and the weights form an entry in $Lc$. The reception field will slide along $c$ with a stride of 1. We denote by $G(x, a)$ the probability density function of Gaussian distribution with mean zero and standard deviation $a$, then the weight of filter $i$ with parameter $(n_i, a_i)$ assigned to the $j$-th item in the reception field is

$$F_{i,j} = \nu(j; n_i, a_i)/\sqrt{\sum_{r=1}^{d} \nu^2(r; n_i, a_i)} \quad \text{where } \nu(j; n_i, a_i) = \frac{\partial^{n_i}}{\partial x^{n_i}} G(x, a_i)\mid_{x=j} \tag{5}$$

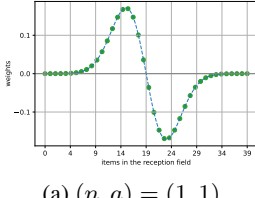 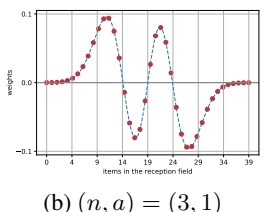 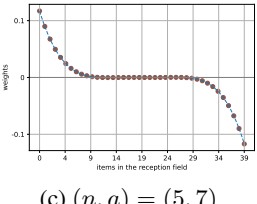

(a) $(n, a) = (1, 1)$          (b) $(n, a) = (3, 1)$          (c) $(n, a) = (5, 7)$

Figure 3: Three DoG filters with the reception field of 40 items.

Thus, these filters can detect group-wise comparison patterns and encode them into $\tanh(\gamma L c)$.

For representing the TGP matrix $L$, we use $\theta$ to denote the parameters $(n_1, a_1, \ldots, n_D, a_D)$ of all $D$ DoG filters used. With our wavelet filter representation approach, the search of an adequate TGP matrix over the entire matrix space degenerates to finding the optimal $\theta$, which saves the computation cost greatly and makes it not necessarily dependent on the size of $c$.

## 4.2 PARAMETER OPTIMIZATION

Algorithm 1 shows the details of our approach. Specifically, at each round $t$, we explore the loss function space by creating $M$ trajectories, each trajectory with a set of DoG filter parameter $\mu^m$ sampled independently from a truncated normal distribution $\mathcal{N}_{\text{truncated}}(\theta_{t-1}, \sigma^2 I)$. The truncated normal distribution has support on $[0, +\infty)$. We denote the probability density function of $\mu \sim \mathcal{N}_{\text{truncated}}(\theta, \sigma^2 I)$ as $\pi(\mu \mid \theta)$. For each loss function $\ell^m$, the prediction model is then trained to minimizes $\ell^m$ using stochastic gradient descent (SGD) and the learned model $f_{\hat{w}}$ makes predictions on the validation set $\{\hat{c}^{(i)} : i \in V\}$. We evaluate the current policy by calculating the negative total SPO loss of $f_{\hat{w}}$ on the validation set $R_t^m = -\sum_{i \in V} \ell_{\text{SPO}}(\hat{c}^{(i)}, c^{(i)})$.

The sampling policy can be updated with commonly used reinforcement learning algorithms. Two classic algorithms are listed here in our work, and other policy updating algorithms are also applicable. The first one we have used is REINFORCE algorithm (Williams, 1992)

$$\theta_t \leftarrow \theta_{t-1} + \frac{\eta}{M} \sum_{m=1}^{M} [R_t^m \nabla_\theta \log \pi(\mu^m \mid \theta_{t-1})] \tag{6}$$

Besides, the PPO2 algorithm (Schulman et al., 2017) has also been applied:

$$\theta_t \leftarrow \arg\max_\theta \frac{1}{M} \sum_{m=1}^{M} \min \left( \frac{\pi(\mu^m \mid \theta)}{\pi(\mu^m \mid \theta_{t-1})} R_t^m, \text{CLIP}\left( \frac{\pi(\mu^m \mid \theta)}{\pi(\mu^m \mid \theta_{t-1})}, 1 - \epsilon, 1 + \epsilon \right) R_t^m \right) \tag{7}$$

---

**Algorithm 1:** Automatic Prediction and Optimization Connector (APOC)

**Input:** gradient-based prediction model $f_w$, initial loss parameters $\theta_0$, number of search iterations $T$, number of samples per iteration $M$, exploration ratio $\sigma$;
**Output:** final loss parameters $\theta^*$, final prediction model parameters $w^*$;

1 **for** *each* $t \in [1, T]$ **do**
2     **for** *each* $m \in [1, M]$ **do**
3        Sample DoG filter parameters $\mu^m$ from the truncated normal distribution $\mathcal{N}_{\text{truncated}}(\theta_{t-1}, \sigma^2 I)$;
4        Compute the ATGP loss function $\ell^m$ which is parameterized by $\mu^m$ Train the prediction model
          $\hat{w} = \arg\min_w \sum_{i \in S} \ell^m(f_w(x^{(i)}), c^{(i)})$;
5        Evaluate the model on the validation set $\hat{c}^{(i)} = f_{\hat{w}}(x^{(i)})$ for all $i \in V$;
6        Compute the reward $R_t^m = -\sum_{i \in V} \ell_{\text{SPO}}(\hat{c}^{(i)}, c^{(i)})$;
7     Update $\theta_t$ according to REINFORCE (6) or PPO2(7);
8 $\theta^* = \theta_T$, $w^* = \arg\min_w \sum_{i \in S} \ell^*(f_w(x^{(i)}), c^{(i)})$ where $\ell^*$ is the ATGP loss parameterized by $\theta^*$.

---

Following the common practice in policy gradient algorithms, we subtract a baseline value from $R_t^m$ to reduce the variance of the gradient estimator (Schulman et al., 2015). Here we use the averaged reward from all $M$ trajectories as the baseline value.

## 5 EXPERIMENTS

The proposed APOC method is applied to three predict-then-optimize problems with real-world datasets to validate its performance. Those three PTO problems contain optimization problems with different representative combinatorial structures, *i.e.,* ranking, knapsack, and shortest path problem. Empirically, our experimental results demonstrate that significant performance gains can be achieved against some previously proposed algorithms by finding a better loss for the prediction phase.

### 5.1 DATASET DESCRIPTION

**Energy Cost-Aware Scheduling**   This dataset contains two years of historical energy price data from the day-head market of the Irish Single Electricity Market Operator (SEM-O). The energy prices for each half-hour of a day need to be predicted with some given feature data (Simonis et al.) including estimates of the temperature, wind speed, $CO_2$ intensity, etc. We need to select a subset of 48 half-hours in one day based on the predicted prices to schedule the machines to minimize the total expenditure of the day. Here we formulate the optimization problem as a top-$k$ *ranking problem*, where the order of the top-$k$ half-hours in a day determines the minimal total expenditure of the day. We evaluate algorithms with $k \in \{5, 10, 15\}$. The data set contains 37,872 instances.

**Knapsack Formulation of Portfolio Optimization**   We use the historical daily price and volume data from 2004 to 2017 from partition SP500 of the Quandl WIKI dataset (QUANDL, 2020). The task is to predict the future returns for a set of 505 candidate stocks, and generate daily portfolios based on the predictions. The optimization problem is formulated as a *knapsack problem* in which the decision maker needs to choose stocks based on estimated returns to maximize the total expected reward under the fixed budget. The data set provides 150 instances.

**Shortest Path**   We use the processed data from Mandi & Guns (2020) where the goal is to find the shortest path of the given source-destination pairs on a Twitter ego network (McAuley & Leskovec, 2012). The network consists of 231 nodes and 2861 edges where the edge weights need to be predicted using node and edge features, with 115 instances in total.

### 5.2 EXPERIMENTAL SETTINGS

**Baseline methods**   We compare with the follow PTO methods. $l_2$ distance: two-stage method with conventional $l_2$ regression. QPTL/IntOpt: exerts quadratic/log-barrier regularization on the solution parameters to obtain non-zero gradient for differentiable training. SPO+: uses continuous convex surrogate of the SPO loss function for sub-gradient based training. NeuralSort: differentiably relaxing the permutation matrix in sorting problems. DiffOpt: a recent method which approximates the gradient by the solution difference with perturbed prediction parameters. The outlines of these methods are discussed in Section 2.

**Experiment settings**   For each experiment, we use the same prediction model architecture and train/validation/test splits across different methods for fair comparisons. We use Adam optimizer for parameter training across all experiments, where hyperparameters (learning rate, ATGP temperature $\gamma$) are obtained by grid search using the validation performance. For Energy Cost-Aware Scheduling, the prediction model is instantiated to be a 4-layer neural network (hidden layer sizes of 500-300 with ReLU as activation function) with learning rate $5 \times 10^{-5}$ and $\gamma = 5.0$. For Shortest Path, the prediction model is a 4-layer network (hidden layer sizes of 100-100 with ReLU as activation function) with learning rate $5 \times 10^{-4}$ and $\gamma = 3.0$. For Portfolio Optimization, we use a 5-layer neural network (hidden layer sizes of 100-100-100 with ReLU as activation function) with learning rate $5 \times 10^{-5}$, $\gamma = 3.0$. We use Gurobi optimizer to solve all combinatorial optimization problems and use CVXPY (Diamond & Boyd, 2016; Agrawal et al., 2018) and QPTH (Amos & Kolter, 2017) for differentiable optimization.

### 5.3 RESULTS

This subsection shows the evaluated performance of our algorithm on the three PTO problems described above. The goal is three-fold: 1) to demonstrate that our algorithm outperforms the previous proposed algorithms by a significant margin; 2) to analyze the sensitivity of our algorithm to the hyperparameters; 3) to validate the effectiveness of our proposed DoG filters by an ablation study.

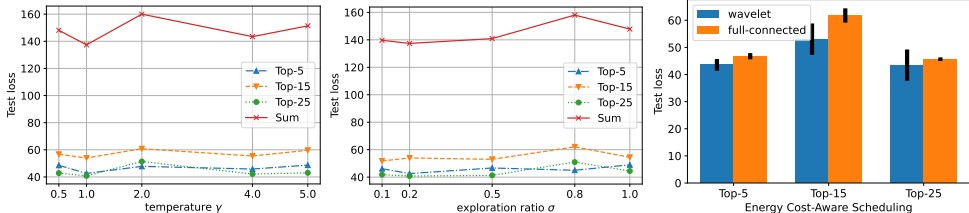

Figure 4: Left and mid panel show the performance of Algorithm 1 on Energy Cost-Aware Scheduling problem with different $\gamma$ (fixing $\sigma = 0.2$) and different $\sigma$ (fixing $\gamma = 1.0$), respectively. The right panel shows the ablation study on the DoG wavelet filter parameterization ($\gamma = 1.0$, $\sigma = 0.2$), with black segments showing the (5%, 95%) percentile interval.

**Performance Comparison**    The results in Table 1 show that our proposed APOC algorithm yields significantly lower SPO losses than other algorithms on all three PTO problems. For Energy Cost-Aware Scheduling and shortest path problem, the ATGP loss function searched by REINFORCE algorithm induces better predictions while PPO2 algorithm is better on knapsack problem.

Table 1: Comparison of SPO losses with baseline algorithms on the three PTO problems. "–" means either the algorithm is inapplicable or cannot achieve a result in satisfying range with our best efforts.

| Algorithm | Energy Cost-Aware Scheduling | | | Knapsack | Shortest Path |
|---|---|---|---|---|---|
| | top-5 | top-15 | top-25 | | |
| $l_2$ distance | 53.84 | 80.75 | 51.17 | 26.57 | 223 |
| QPTL | 68.86 | 110.34 | 74.31 | 27.14 | 197 |
| IntOpt | – | – | – | 27.06 | 92 |
| SPO+ | 47.27 | 65.26 | 44.93 | 26.03 | 138 |
| NeuralSort | 52.69 | 73.23 | 45.43 | – | – |
| DiffOpt | 50.28 | 71.86 | 57.06 | 27.21 | – |
| APOC-REINFORCE | **42.59** | **53.97** | **40.81** | 25.70 | **63.87** |
| APOC-PPO2 | 49.50 | 66.80 | 55.56 | **24.93** | 78.03 |

**Hyperparameter sensitivity**    In our algorithm, there are two key hyperparameters: the temperature parameter $\gamma \in (0, +\infty)$ and the exploration ratio $\sigma \in (0, +\infty)$. Each hyperparameter stands for a trade-off in the searching process. Lower temperature $\gamma$ gives a smoother ATGP loss which is beneficial to the training of the prediction model but as Lemma 1 states, the ATGP loss with higher $\gamma$ approximates the true TGP loss closer. The exploration ratio $\sigma$ balances a classic exploration-exploitation trade-off in the searching algorithm. Sampling policy with a smaller $\sigma$ *exploits* the feedback more from the last step while the policy with larger $\sigma$ *explores* a larger area of the parameter space. We present the influences of $\gamma$ and $\sigma$ on our searching algorithm in Figure 4. The results show that the proposed method is generally robust and insensitive to these hyperparameters.

**Ablation Study**    To understand the contribution of the DoG wavelet filter, we conduct an ablation study that compares the performance of our APOC algorithm with the searching algorithm in which the TGP matrix is parameterized by a fully connected dense layer. As is shown in right panel of Figure 4, the wavelet filter yields a significant decrease in the SPO loss on the test set.

## 6 CONCLUSION

To tackle the disalignment between prediction loss and optimization goal in PTO problems, we propose to use the differentiable ATGP loss which captures the total group preorder. A corresponding APOC method is further proposed to automatically search for a good surrogate loss with wavelet filters to capture the group-wise comparison of different sizes and weights. The method is well validated on three different PTO problems and shows great potential for more PTO problems. Starting from the limitations of this work, the following directions are expected for future work: 1) extend the scope of APOC framework to PTO problems beyond the ones with strong ranking property, 2) explore how to reduce the searching time of the iterations within APOC framework.

## REPRODUCIBILITY STATEMENT

**Source code**   We have made the source code for the proposed APOC available in the following repository: `https://github.com/Microsoft/AutoPredOptConnector`.

**Proof of *Lemma 1* and *Theorem 1***   We have attached the complete proof of *Lemma 1* and *Theorem 1* in the Appendix.

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

APPENDIX

## 1   PROOF OF LEMMA 1

*Proof.* Notice that the sign function is the derivative of a "V"-shape function:

$$\text{sign}(x) = \mathbf{1}_{x>0} - \mathbf{1}_{x<0}$$

$$= \begin{bmatrix} 1 \\ 0 \end{bmatrix}^{\top} \arg\max(x, 0) - \begin{bmatrix} 1 \\ 0 \end{bmatrix}^{\top} \arg\max(-x, 0)$$

where $\arg\max(x, 0) = [1, 0]^{\top}$ if $x \geq 0$ and $[0, 1]^{\top}$ otherwise. We can use softmax function to approximate the argmax, *i.e.,*

$$\arg\max(x, 0) = \lim_{\gamma \to +\infty} \begin{bmatrix} \frac{e^{2\gamma x}}{e^{2\gamma x}+1} \\ \frac{1}{e^{2\gamma x}+1} \end{bmatrix}$$

and

$$\begin{bmatrix} 1 \\ 0 \end{bmatrix}^{\top} \left( \begin{bmatrix} \frac{e^{2\gamma x}}{e^{2\gamma x}+1} \\ \frac{1}{e^{2\gamma x}+1} \end{bmatrix} - \begin{bmatrix} \frac{e^{-2\gamma x}}{e^{-2\gamma x}+1} \\ \frac{1}{e^{-2\gamma x}+1} \end{bmatrix} \right) = \frac{e^{2\gamma x} - 1}{e^{2\gamma x} + 1}$$

$$= \tanh(\gamma x)$$

Combined all equalities together, we have $\text{sign}(x) = \lim_{\gamma \to +\infty} \tanh(\gamma x)$. $\square$

## 2   PROOF OF THEOREM 1

**Notations.** For a matrix $A$, $A_i$ refers to its $i$-th row and $A_{:,i}$ refers to its $i$-th column. $\mathcal{L}(A)$ denotes the column space of $A$ and $P_A$ is the orthogonal projection matrix onto $\mathcal{L}(A)$. $A^+$ denotes the pseudo-inverse of $A$; $||A||_F = \sqrt{\sum_i ||A_i||_2^2}$ is the Frobenius norm of $A$.

For convenience of introducing our proof, we describe our data model and assumption in matrix notations. Let $N = |S|$ be the number of samples, we can row-stack the feature vectors to be matrix $X \in \mathbb{R}^{N \times k}$. Also let the ground truth linear regression parameter matrix be $H^* \in \mathbb{R}^{k \times d}$ ($H^*$ is transposed here compared to the main text since we're viewing the feature/response variables as row vectors instead of column vectors), so the row-stack matrix of ground truth response values is $C^* = XH^*$. We observe $C = C^* + E$ where $E$ is the noise matrix and the noise level is defined to be $||E||_F^2$.

For a certain loss metric $l$, the minimizer is denoted by

$$\hat{H}^l = \arg\min_H l(XH, C)$$

and the prediction is denoted by

$$\hat{C}^l = X\hat{H}^l$$

Any specific optimization problem with strong ranking property corresponds to a subset of its total group pre-orders. We formalize this with the following definition.

**Definition 6.** *A function $f$ is called a* sub-order function *if it maps a vector to a subset of its total group pre-orders. We will also call $f(a)$ the sub-orders of $a$ induced by $f$.*

Now a sub-order function $f$ would specify the optimization problem at hand. Each sub-order function $f$ would also correspond to a TGP matrix whose rows compares the sub-groups induced by $f$; we denote this matrix by $L_f$. Take Ranking problems for example, $f(a) = f_{\text{Ranking}}(a)$ would map $a$ to the subset of $\text{TGP}(a)$ where the elements $(u_1, u_2)$ satisfy $|u_1| = |u_2| = 1$, and $L_f$ would be a matrix that maps a vector to another vector of its pairwise differences.

Now for sample matrix $A$ and label matrix $B$, the two loss metrics whose robustness to noise we aim to compare are:

$$\ell_{l_2}(A, B) = ||A - B||_F^2, \quad \ell_{\text{ATGP}}(A, B) = ||\tanh(\gamma \cdot L_f A) - \text{sign}(L_f B)||_F^2$$

Given a sub-order function $f$ and a vector $a$, we define $\Delta_f(a)$ to be the minimum perturbation on $a$ to change $a$'s sub-orders induced by $f$; similarly for a matrix $A$ which consists of a row-stack of samples.

**Definition 7.** *For a vector $a$ or a matrix $A$,*

$$\Delta_f(a) = \min_e ||e||_2^2 \text{ s.t. } f(a) \neq f(a+e)$$
$$\Delta_f(A) = \min_j \Delta_f(A_j)$$

(8)

Let us reformulate the assumption for Theorem 1 with the above notations.

**Assumption 1** (Reformulated)**.** *The feature matrix $X$ has a low rank structure (meaning the samples belong to a linear subspace of less than $k$-dimension) except for the sample indexed by $g$ which can induce different sub-orders with minimum perturbation, i.e., $\Delta_f(C_g^*) = \Delta_f(C^*)$.*

**Validity and Intuition of Assumption 1**: it is easy to construct data models which satisfies this assumption; details at the end of the document. This assumption says that the sample which is most likely a outlier in the dataset does not share the low-rank structure of other samples. Considering the low-rank structure of most real world datasets and the nature of outliers, this assumption is very likely to hold intuitively. We also believe our conclusion can be extended to non-linear settings where the low-rank linear subspace in this assumption is replaced by low-dimensional manifolds.

**Proof for Theorem 1.** We're now ready to present the proof of Theorem 1, which is an immediate consequence of Lemma 2 and Lemma 3 below.

**Lemma 2.** *Under Assumption 1, we have*

$$\mathcal{R}(\ell_{l_2}) \leq \Delta_f(C^*)$$

*Proof.* For $l_2$ loss the solution is given by

$$\hat{H}^{l_2} = X^+(C^* + E) = X^+C^* + X^+E = H^* + X^+E$$

(9)

and thus

$$X\hat{H}^{l_2} = C^* + XX^+E = C^* + P_X E$$

(10)

Let $U \in \mathbb{R}^{N \times k}$ be a orthogonal basis for $\mathcal{L}(X)$, then $P_X = UU^T$. Let a orthogonal basis of the complement of $\mathcal{L}(X)$ be $U^\perp$. $[U; U^\perp] = U_{full}$ then forms an orthonormal basis of $\mathbb{R}^N$.

Let a normal vector of the subspace in Assumption 1 be $\bar{\alpha} \in \mathbb{R}^k$ and let the index for the special sample be $g$. Then $\Delta_f(C^*) = \Delta_f(C_g^*)$ and also $\bar{\alpha}^T x_i = 0$ for $i \neq g$, $\bar{\alpha}^T x_g \neq 0$. Now, we can scale $\bar{\alpha}$ to $\alpha$ s.t. $\alpha^T x_g = 1$. Therefore we now have

$$X\alpha = e_g$$

where $e_g$ is the $g$-th standard basis (one-hot) vector. Now $e_g \in \mathcal{L}(X) = \mathcal{L}(U)$ and thus basis vectors in $U^\perp$ are all perpendicular to $e_g$. This means the $g$-th entry of any basis vector in $U^\perp$ is 0. Since any row of $U_{full}$ has unit norm, the $g$-th row of $U$ must have unit norm, which also indicates that the $g$-th row of $P_X = UU^T$ has unit norm. Let the $g$-th row of $P_X$ be $p \in \mathbb{R}^n$.

Now, we can find a matrix $E$ s.t. $P_X E$ changes the sub-orders exactly in the $g$-th row of $C^*$. The noise level of such a matrix $E$ would then be an upper bound on $\mathcal{R}(\ell_{l_2})$. Denote $c^* = C_g^*$. Let $(i,j)$ be the pair in $f(c^*)$ with smallest $|c_i^* - c_j^*|$. We simply need to find $E_{:,i}, E_{:,j} \in \mathbb{R}^N$ with minimum $||E_{:,i}||_2^2 + ||E_{:,j}||_2^2$ s.t. $c_i^* + p^T E_{:,i} = c_j^* + p^T E_{:,j}$. It can be easily seen that the optimal solution is given by

$$E_{:,i}^* = \frac{c_i^* - c_j^*}{2} \cdot p; E_{:,j}^* = \frac{c_j^* - c_i^*}{2} \cdot p$$

and thus the minimum noise level is

$$||E_{:,i}^*||_2^2 + ||E_{:,j}^*||_2^2 = \frac{(c_i^* - c_j^*)^2}{2} = \Delta_f(C_g^*) = \Delta_f(C^*) \geq \mathcal{R}(\ell_{l_2})$$

by which our proof is done. $\square$

**Lemma 3.** $\Delta_f(C^*) \leq \mathcal{R}(\ell_{ATGP})$.

*Proof.* The proof of this lemma is rather simple; recall that the ATGP loss seeks to find $H$ which minimizes $\ell_{\text{ATGP}}(XH, C^* + E) = ||\tanh(\gamma \cdot L_f X H) - \text{sign}(L_f(C^* + E))||_F^2$. Now as long as $E$ doesn't change the sub-orders in $C^*$'s rows, $\text{sign}(L_f(C^* + E)) = \text{sign}(L_f C^*)$ so $\ell_{\text{ATGP}}$ remains intact under the noise $E$, and thus it would still give the optimal solution. Therefore the minimum noise level which changes the sub-order in at least one of $C^*$'s rows would always be a lower bound for $\mathcal{R}(\ell_{\text{ATGP}})$. □

This bound should have a large gap with the best bound since the decision boundary usually shouldn't change a lot even when we corrupt a portion of the samples. Here we only corrupt one sample.

**Example of constructing data that satisfies Assumption 1**. Generating $X$ which satisfies the requirement is straightforward. Now let $H_1, H_2 \in \mathbb{R}^d$ be nearly orthogonal vectors of $x_g$. Randomly generate $H_3, ..., H_n$ and flip their signs s.t. $k-1$ out of them have $H_j^T x_g < 0$ the rest have $H_j^T x_g > 0$. This ensures that $H_1^T x_g, H_2^T x_g$ are the $k$-th and $k+1$-th biggest items in $C_{g,:}^*$. By genericity, they are the closest pair in all pairs of entries of $C^*$ and therefore $\Delta_k(C^*) = \Delta_k(C_g^*)$.

