# OpenReview forum: "Automatic Loss Function Search for Predict-Then-Optimize Problems with Strong Ranking Property"
_ICLR.cc/2022/Conference — ICLR 2022 Poster_

### Official Review · Reviewer_R87E · 2021-10-22

**Correctness:** 4
**Technical Novelty And Significance:** 3
**Empirical Novelty And Significance:** 4
**Recommendation:** 6
**Confidence:** 4

**Main Review:**

Here are some of the paper’s strengths and weaknesses:

(+) I thought that the motivation and the illustration with Figure 2 was compelling. Combinatorial optimization problems can be very sensitive: perturbing the problem’s parameters by just a little bit can cause the solution to change by a lot. Therefore, naively optimizing a standard loss function to predict the problem’s parameters can lead to poor downstream results for the optimization task.

(+/-) I think that the “strong ranking property” is promising, but it could be fleshed out. When it is first described in Section 1, it’s very abstract and hard to grasp what it means. The ranking example in Section 3.1 helps, but I think that it comes too late. Moreover, it’s never really explained why certain combinatorial problems have this property, so the reader has to trust that it’s a useful property and that the problems in the experiments section satisfy the property. A few fleshed-out examples and explanations would be helpful.

(-) The theoretical results leave something to be desired, since they show that for a very specific example, optimizing using the proposed approach is weakly better than optimizing using the l2 loss. This would, of course, be more compelling if the authors could provide an example where the proposed approach is better than using the l2 loss by some non-zero margin.

(-) The applied approach to representing $L$ using discretized Derivative-of-Gaussian filters seemed a bit ad hoc and I didn’t really understand the deeper intuition behind why this might be a good approach.

More detailed comments:
- Figure 2: It was a little hard to tell where the stars and triangles lie on the x-y axes.
- Definition 3: At first, it was a bit hard to grasp what L is supposed to represent. Maybe this could be explained earlier, and with an example.
- Equation (3): Should sign be tanh?

**Summary Of The Paper:**

This paper studies a problem where the goal is to solve an optimization problem, but key parameters of the optimization problem (like a road network’s edge weights) are unknown and can only be predicted from historical data. This line of research is called “predict then optimize.” For the prediction step, the first thing one might try would be to minimize a standard loss function like the l2 loss, but this loss function may not necessarily lead the downstream optimization problem to return the highest quality solutions. This paper studies a way to learn a good loss function for the downstream optimization problem at hand.

To give an example, if the optimization problem is to rank a set of elements, even if one can predict the elements’ values with low l2-error, the resulting ranking might be totally incorrect.
Motivated by this example, this paper’s results apply to combinatorial optimization problems that satisfy a “strong ranking property,” which essentially means that the optimization problem’s solution only depends on group-wise comparisons of the problem’s parameters (rather than on the specific values of the parameters themselves).

Let $c \in R^d$ be the true parameters and $\hat{c}$ be the learned parameters. The authors propose that the loss function one should optimize should ideally have the form $||sign(L\hat{c}) - sign(Lc)||$, where $L$ is a matrix of size ${2^d \choose 2} \times d$ that allows for all $2^d \choose 2$ group-wise comparisons between all groups of parameters. Since this ideal loss function is computationally intractable, the authors propose replace $sign$ with $\tanh$. Then, they propose fixing the number of rows of $L$ and parameterizing the matrix using a set of discretized Derivative-of-Gaussian filters, which they optimize using the reinforcement learning algorithm REINFORCE [Williams ‘92] (the details here are a bit unclear to me).

They evaluate their algorithm for three problems: a scheduling problem, portfolio optimization, and shortest path. They compare against a number of different predict-then-optimize algorithms from prior research and show that their algorithm has much better losses on the problems studied.

**Summary Of The Review:**

Overall, my recommendation is “weak accept” because I think the motivation is compelling and the “strong ranking property” seems to open up this predict-then-optimize framework to important combinatorial optimization problems (though I’m not very familiar with prior research on this specific topic). As I mentioned in the main review, there are a lot of ways the paper could be fleshed out to increase it’s potential impact.

---

> ### Author Response · Authors · 2021-11-16
> **Reply to Reviewer R87E**
>
> We thank the reviewer for the valuable comments and will update our paper according to the suggestions. Our responses to the questions are as follows:
> ## Discussion on "Strong Ranking Property"
> We thank the reviewer for the good suggestion.
>  Actually the scope of the combinatorial problems that have the  "strong ranking property" can be indicated from formula (1) in the paper. We can only deal with the case where the predicted "values" of the elements in the problem are independent from the feasibility of the problem. Therefore, not only the three combinatorial optimization problems we have experimented in our paper are in this scope, other problems such as the bin packing problem with predicted values of items is also within this scope. Moreover, more explanation about the scope of our investigated problems can refer to the section "**Regarding Definition 1**" in our reply to [Reviewer *8c8S*](#note_9U-8pfeFpZ1).  We will also add more details about the problem scope in our revised paper to make it easier for readers to understand.
>
> ## Theoretical Results
>  One major challenge of the theretical proof of the TGP loss is that there is no analytical solution to this loss, thus it is hard to bound the performance of this loss under certian noise level. However, even starting from the simple example in Figure 2, it is quite intuitive that $l_2$ loss is not ideal in most combinatorial optimization problems, and this example itself shows a non-zero margin of advantage of the TGP loss over the $l_2$ loss. Despite of the difficulty of investigating the property of the TGP loss in theory, hopefully our proof can inspire more theoretical development towards this problem.
>
>
>
> ## Intuition of DoG Filters
>
>
> For some combinatorial optimization problems, the solver algorithm needs **group-wise** comparisons within the items [1]. For example in shortest path problem with unknown edge weights, the solver needs to compare the length of different paths, which is the sum of a group of edge weights. By allowing the entries in $L$ vary between $[-1, 1]$, the linear transformation associated with $L$ will map the parameter vector $c$ to an embedding vector whose entries are the differences between weighted sums of different groups. For example, $L=[1,1,−1]$, where the i-th entry of $L$ ($L[i]$) represents the value of the i-th item, will yield the difference between group A consisting of item 1 and 2, and the group B of item 3.
>
> We use DoG filters to parameterize the entries in $L$, to avoid the optimization issues due to the large number of free entries in $L$. Each DoG filter will specify a row in $L$ and each DoG filter helps to extract a **group-wise** comparison by applying the inner product with the parameter $c$, where such group-wise comparisons correspond to the order of subsets in the definition of TGP.
>
>
> ## More Detailed Comments
>  ### Symbol in Figure 2
> We will use more obvious shapes to distinguish the symbols in Figure 2 in our updated version of paper, and we thank the reviewer for the good suggestion.
>
> ### $L$ in Definition 3
> Consider the 3-item ranking problem in Section 3.2. A 3-by-3 matrix
> $L = \left(\begin{matrix}-1 & 0 & 1 \\\ 0 & -1 & 1 \\\ 1 & -1 & 0 \end{matrix}\right)$ is an example of good TGP matrices for this problem. The linear transformation associated with $L$ maps the parameter vector $c$ to an embedding vector $e = [c_3 - c_1, c_3 - c_2, c_2 - c_1]$. This embedding passed into the point-wise sign function will demonstrate the **pair-wise** comparisons in the parameter $c$, which meets the requirement of a ranking algorithm. By fitting DoG filters in the matrix $L$, it can flexibly represent **group-wise** comparisons which make up the orders of the subsets in the TGP definition.
>
> ### "sign" Function in Equation (3)
> The sign function here is not a typo. Compared to Eq (2), we only approximate the first sign function with the tanh function because that is where the non-differentiablity comes from. For the second term in Eq (3), it is directly calculated as a constant, thus no need to be approximated.
>
> ---
> ### References
> [1] Khalil, E., Dai, H., Zhang, Y., Dilkina, B., & Song, L. "Learning Combinatorial Optimization Algorithms over Graphs." *Advances in Neural Information Processing Systems* (2017).

---

### Official Review · Reviewer_AESs · 2021-11-03

**Correctness:** 3
**Technical Novelty And Significance:** 3
**Empirical Novelty And Significance:** 3
**Recommendation:** 8
**Confidence:** 4

**Main Review:**

Overall, the paper is well-written.
The proposed ideas are reasonable and they are presented in a logical manner.
Performance evaluation based on three examples - energy cost-aware scheduling based on historical energy price data, portfolio optimization based on SP500 data, and shortest path finding in Twitter ego network - demonstrate that the proposed ATGP loss and the APOC  (Automatic Prediction and Optimization Connector) algorithm may have potential advantages over other loss functions and algorithms.

One major concern regarding the paper is that the authors do not consider the rich literature on optimal decision making - especially, decision making under uncertainty - that has a very long history in operations research and other fields.
For example, the SPO loss shown on page 3 is conceptually identical to what is known as "regret" in decision theory.
Regret estimates the expected performance loss due to using a suboptimal predictor (or operator) instead of the optimal predictor (or operator), typically due to the fact that the true model parameters or the optimal predictor/operator are unknown (e.g., due to aleatoric and/or epistemic uncertainties).

Furthermore, if the predictor (or operator) is optimized to minimize the regret (or the SPO loss shown on page 3) over a class of possible predictors/operators, that leads to the so-called MOCU (mean objective cost of uncertainty) that quantifies the operational cost induced by model uncertainties.
Both regret and MOCU have been widely studied in problems such as optimal decision making, optimal experimental design, and active learning - especially in the presence of uncertainty.
However, the authors, unfortunately, do not discuss the relation of the presented work with the existing studies based on regret or MOCU.

Another concern is the limited performance validation of the proposed approach presented in the current paper.
For example, it is unclear whether the ablation study in Figure 4 clearly shows the advantage of the DoG wavelet for automatic search of ATGP, as the improvement does not seem to be significant when compared to the fully-connected network.
The results in Table 1 also show that the performance of the ATGP via APOC significantly depends on the choice of the optimization scheme (e.g., REINFORCE vs PPO2), since in most cases, not both schemes outperform the other alternatives.

Additional performance assessment results are needed to further validate the advantages of the proposed scheme and the above points should be clearly discussed with mentions about the potential limitations of the current method.


**Summary Of The Paper:**

In this paper, the authors propose a novel loss function called the approximated total group preorder (ATGP) loss function, which aims to address the potential limitations of l2 loss in predict-then-optimize (PTO) problems.
The authors argue that the use of l2 loss in the prediction phase is "misaligned" with the final goal when the ultimate objective is to make discrete decisions in a combinatorial optimization setting.
While the so-called SPO (smart PTO) loss, recently proposed by Elmachtoub and Grigas (2021) provides a remedy, the authors note that it cannot be used with gradient-based learning algorithms.
The authors aim to rectify this issue by proposing a novel loss function called the total group preorder (TGP) loss - for better alignment - and then an algorithm for automatically searching for an approximate TGP (ATGP) loss function so that it is differentiable and therefore can take advantage of efficient gradient-based learning schemes.

**Summary Of The Review:**

The paper deals with an interesting problem and the proposed method is well-motivated and presented in a logical manner.
The experimental results based on three different datasets show that the proposed method may have potential advantages over other existing schemes.
However, despite the conceptual similarity between SPO loss (which the TGP/ATGP tried to improve) and regret/MOCU in decision theory, the authors do not discuss the relation of their work to the rich existing literature on regret and MOCU.
Furthermore, additional performance evaluation results are needed to clearly demonstrate the robustness and benefits of the proposed scheme.
Including discussions on potential limitations of the current method (if any) would strengthen the paper.

---

The evaluation has been updated after reviewing the authors' response.

---

> ### Author Response · Authors · 2021-11-16
> **Reply to Reviewer AESs**
>
> We thank the reviewer for the valuable comments and will update our paper according to the suggestions, especially we will add the discussion on the limitation of this work. Our responses to the questions are as follows:
> ## 1. Relation to Decision Making Under Uncertainty and MOCU
> The SPO loss, defined as the difference between the objective value of an optimal solution and that of a potential sub-optimal solution, is so-called **regret** in decision theory [1].
>     When the potential sub-optimal solution is a robust solution with respect to the uncertainty of parameters in the optimization problem, the SPO loss is **mean objective cost of uncertainty** (MOCU) [2, 3, 4].
>     Compared to MOCU, APOC chooses the sub-optimal solution by solving the optimization problem with the predicted parameters.
>     Besides, MOCU utilizes a Bayesian perspective by assuming some prior distribution on the unknown parameters while our proposed APOC method makes no such assumption. We will add the review of this stream of research in the updated version of our paper.
>
> ## 2. Significance in Ablation Study
> The performance gains from DoG filters are statistically significant in Top-5 and Top-15 rankings (the black lines in Figure 4 show the (5\%, 95\%) percentile interval). For the Top-25 ranking (the number of all items is 48), although the gain seems to be a bit less significant in this single setting (compared to Top-5 and Top-15 settings), the mean loss is lower with DoG filters than with fully-connected layer. Actually the performance gain in this ranking problem is generally decreasing with the increasing number of the items to select. For example, if we select top 50 out of 50 items in the extreme case, the performance gain of our method would be 0 compared to even random selection. Therefore, the significance of performance is supposed to get weaker as the number of items to be selected increases.
>
> Moreover, the fully-connected layer in this ablation study is also used to learn an ATGP loss, which is consistent with the idea we propose in this paper. The results in Table 1 can well show the performance gain of using ATGP loss, and by using the DoG filters in our APOC method, we can achieve a better result compared to pure fitting with fully-connected layer.
>
> ## 3. Significance over Other Methods and Choice of REINFORCE and PPO2
> It is shown in Table 1 that APOC-REINFORCE outperforms all the other methods in all the cases we consider in the experiments. Although APOC-PPO2 does not always beat other methods in all cases, it can achieve the best performance in the knapsack problem.
>
> We list these two methods with different optimization schemes to show that our method can actually use different optimization schemes in the training procedure.  The optimization scheme is in itself not a settled research area and still needs further research on which scheme works better under what circumstances. Further explanation of the optimization methods can refer to the section "**More details for Algorithm 1 - a, b**" in our reply to Reviewer *8c8S*.
>
>
> ---
> ### References
> [1] Bell, David E. "Regret in decision making under uncertainty." *Operations research* 30.5 (1982): 961-981.
>
> [2] Yoon, Byung-Jun, Xiaoning Qian, and Edward R. Dougherty. "Quantifying the objective cost of uncertainty in complex dynamical systems." *IEEE Transactions on Signal Processing* 61.9 (2013): 2256-2266.
>
> [3] Boluki, Shahin, Xiaoning Qian, and Edward R. Dougherty. "Experimental design via generalized mean objective cost of uncertainty." *IEEE Access*  7 (2018): 2223-2230.
>
> [4] Guang Zhao, Edward Dougherty, Byung-Jun Yoon, Francis J. Alexander, Xiaoning Qian. "Bayesian active learning by soft mean objective cost of uncertainty." *International Conference on Artificial Intelligence and Statistics*. PMLR, 2021.

---

> > ### Comment · Reviewer_AESs · 2021-11-18
> > **Reply to the authors**
> >
> > Thank you for the detailed response to the review comments. It has addressed several doubts/concerns I've raised in the original review. As the authors agreed, discussing the limitations of the current study would be helpful. Furthermore, I suggest including additional performance evaluation results to support the significance of the proposed work in comparison to other methods (and potentially also to better understand the impact of selecting different optimization schemes on the final performance and its sensitivity w.r.t the choice that has been made).

---

> > > ### Author Response · Authors · 2021-11-19
> > > **Reply to Reviewer  AESs**
> > >
> > > We greatly thank you for the reply and really appreciate these helpful suggestions!  We will consider these aspects in our updated version.

---

### Official Review · Reviewer_8c8S · 2021-11-03

**Correctness:** 3
**Technical Novelty And Significance:** 3
**Empirical Novelty And Significance:** 3
**Recommendation:** 6
**Confidence:** 4

**Main Review:**

My main comments are listed as follows:

1.	Regarding Definition 1 (Total group preorder): TGP© defines all pairs that satisfy the priority order with parameter c. The definition makes sense for the ranking problem. However, how does this definition fit into the shortest path/knapsack problem? That is, most combinatorial problems are constrained, then how to consider this group preorder? If we consider one feasible path as an element, then the number of elements increases exponentially. This is related to the comment below Definition 3. How can the curse of dimensionality be avoided?

2.	More details are needed for Algorithm 1:
a.	What is the intuition behind Equation (6) and (7)?
b.	Are there any connections between these two equations? Under what circumstances should Equation (6) be applied?
c.	What is CLIP in Equation (7)?
d.	In Section 4.1, how to determine D (the number of DoG filters used)?
e.	How to choose \gamma and \sigma? How is \sigma related to the problem itself?

3.	How to avoid the overfitting problem in the TGP/ATGP framework?

4.	In the numerical experiments, Table 1 reports the SPO losses with baseline algorithms on the three PTO problem. Could you also report the computational time for different algorithms?

Minor:

1.	In the proof of Lemma 1, “.” is missing in the last sentence.

2.	In the proof of Theorem 1, to make the notation consistent, I suggest to use A_{i,:} to denote its i-th row.

3.	In the proof of Theorem 1, change the notation for loss function $l$ to $\ell$;

4.	I am a bit confused by the proof of Lemma 2: the first term of X\hat{H}^{l_2} is XX^+ C^*. From the derivation, it gets that XX^+ C^* = XH^*=C^*, which implies that P_X C^* = C^*. If C^* is full rank, then P_X= I. Is it a reasonable conclusion? Why?

5.	The proof of Lemma 3 seems not rigorous. Could you use mathematical equation to prove this conclusion?



**Summary Of The Paper:**

This paper aims to solve combinatorial optimization problems with unknown parameters that need to be predicted from observations. It proposed a total group preorder loss and its differential version approximated total group preorder loss for predict-then-optimize (PTO) problems with strong ranking property. It studied a very interesting problem.

**Summary Of The Review:**

Overall, this is an interesting paper. It may help to understand the paper if adding one or two examples other than ranking (constrained optimization problems such as shortest path/knapsack in the modeling section), to illustrate how the new notion of total group preorder and algorithm can be applied. The proofs need to be written in a clearer way.

---

> ### Author Response · Authors · 2021-11-15
> **Reply to Reviewer 8c8S**
>
> We thank the reviewer for the valuable comments. Our responses to the questions are as follows:
>
> ## 1. Regarding Definition 1
> We would like to further use the following two examples to illustrate how our TGP definition fits into Knapsack/Shortest path problem.
>
> **Knapsack with fixed weights and capacity**
> Consider a knapsack with capacity of 10.
> There are 3 items whose weights are $w_1=3, w_2=4, w_3=5$.
> Let the true values of items are $c_1 = 12, c_2 = 4, c_3 = 5$, which are unknown to the decision maker.
> The values of items are the PTO parameters to be estimated from observations.
> Then $\operatorname{TGP}(c) = \\{(\\{1\\},\\{2\\}), (\\{1\\},\\{3\\}), (\\{3\\},\\{2\\}), (\\{1\\},\\{2,3\\}), (\\{1, 2\\},\\{3\\}), (\\{1, 3\\},\\{2\\})\\}$. Note that actually only the first three subsets are sufficient to determine the optimal solution in this case knowing the capacity and the weights for each item, so TGP would include redundant subsets in most cases, which can be further reduced by learning the necessary subsets through our proposed method to some extent.
>
> **Shortest path**
> Consider an undirected graph $G = (V, E)$ where $V = \\{v_1, v_2, v_3\\}$ and $E = \\{e_{1,2}, e_{2,3}, e_{1,3}\\}$ (this graph is a triangle where $e_{i,j}$ connects node $v_i$ and $v_j$). Let $c_{i,j}$ be the edge weight associated with edge $e_{i,j}$. $\\{c_{i,j}: i\neq j\\}$ are the values that the decision maker needs to predict from the observations. Suppose the true values of $c$ are $c_{1,2} = 3, c_{2,3} = 4$ and $c_{1,3} = 5$,
> then for a PTO problem whose goals is to compute the shortest path between $v_1$ and $v_3$, its $\operatorname{TGP}(c) = \\{(\\{e_{1,2}, e_{2,3}\\}, \\{e_{1,3}\\}),
>     (\\{e_{1,2}, e_{1,3}\\}, \\{e_{2,3}\\}),
>     (\\{e_{1,3}, e_{2,3}\\}, \\{e_{1,2}\\}),
>     (\\{e_{1,3}\\}, \\{e_{1,2}\\}),
>     (\\{e_{2,3}\\}, \\{e_{1,2}\\}),
>     (\\{e_{1,3}\\}, \\{e_{2,3}\\})\\}$
>
> For a general combinatorial optimization problem, the size of $\operatorname{TGP}$ set indeed grows exponentially with the number of parameters. However, knowing the complete TGP set is the sufficient not the necessary condition to solve a specific problem. The curse of dimensionality can be avoided by noticing that the optimal solutions of different optimization problems are determined by different _subsets_ of its TGP, e.g., $\\{(\\{3\\},\\{1\\}),(\\{3\\},\\{2\\}),(\\{2\\},\\{1\\})\\}$ for the ranking problem mentioned in Section 3.2 , and our proposed method can automatically learn the important subsets that contribute most to the solution quality to some extent.
> ## 2. More details for Algorithm 1
> **a**, **b**. These are two well-known policy gradient methods for training the reinforcement learning algorithm, which can both be treated as the repeated application of a policy improvement operator and
> a projection operator [1]. This part in our framework can also be replaced by other policy gradient methods such as MPO [2]. These methods consider different aspects in these two operators and might have better training performance in different applications. Currently there is still no theoretical proof that which method is dominantly best in which specific application/problem.
>
> **c**. The $\operatorname{CLIP}$ function in Equation (7) would clip the first parameter to the interval specified by the latter two parameters, so in Equation (7), if the first parameter (the ratio of the two policy probabilities) is lower than $1-\epsilon$, it will be set to $1-\epsilon$; similarly for the other end. This function has the same definition from the original PPO2 paper (Schulman et al., 2017).
>
> **d**. The number of DoG filters $D$ is a hyperparameter, which is set by conducting grid search with the model's validation performance. The final number of kernels is 30, 15 and 8 for Energy, Portfolio, ShortestPath respectively. Similar to the other hyperparameters such as the kernel size and the number of kernels in CNN, there is no simple guideline on selecting the optimal hyperparameter, and it is a separate research field on hyperparameter optimization [3].
>
> **e**. $\gamma$ and $\sigma$ are two hyperparameters. We recommend to set $\gamma=1.0$ and $\sigma=0.2$ as suggested in our sensitivity experiment section. Here $\gamma$ is the temperature hyperparamter that shapes the steepness of approximating the sign function; $\sigma$ is the exploration ratio that balances the trade-off between greedily choosing best possible parameter and random picking a parameter to escape the saddle point.
> More detailed illustration can be found in the hyperparameter sensitivity paragraph in Section 5.

---

> > ### Author Response · Authors · 2021-11-15
> > **Reply to Reviewer 8c8S (continued)**
> >
> > ## 3. Avoiding overfitting
> > The dataset is divided into training set, validation set and test set, and the final model is chosen to be the one with best performance on validation set, which is one of the standard methods to prevent overfitting. Moreover, the reported result in Table 1 is the performance in test set, which is not exposed to the learned model during training, so the good performance is in itself a proof of good generalization ability of our proposed method without the overfitting trouble.
> >
> >
> > ## 4. Computational time
> > The computational time of our algorithms is listed in the table below. In nature our method is searching the loss by iteratively doing the optimization step, while most of the other algorithms compared in Table 1 are one-off methods without repetitive optimization steps. In this sense, our method is suitable for offline training to find the best loss form with the highest optimization performance in predict-then-optimize problems. After fixing the loss form to be the one found by our method, the inference time is no much difference from other models, but can achieve better performance.
> >
> > | Algorithm | Energy Cost-Aware Scheduling (s) | Knapsack (s) | Shortest Path (s) |
> > |:--------:|:--------:|:--------:|:--------:|
> > | APOC-REINFORCE | 1244  | 1730 | 769 |
> > | APOC-PPO2      | 1205  | 1461 | 705 |
> >
> >
> > ## Minor
> > **1**, **2**, **3**. We greatly thank the reviewer for the advice, and will update these parts accordingly in our updated version of the paper.
> >
> > **4**.
> > For your confusion on Lemma 2: since $C^*=XH^*$, the columns of $C^*$ are all inside the column space of $X$, so $P_X C^*=C^*$. Another way to see this is from the view we took in the proof: $X^+C^*$ is the ground truth regression parameter matrix $H^*$, so $XX^+ C^*=XH^*=C^*$.
> >
> >
> > **5**.
> > For Lemma 3, since the ATGP loss function would first transform the response variables to be vectors indicating their sub-orders (mathematically, it would map the response matrix $Y$ to be $\operatorname{sign}(L_f Y)$, see the Equation at the bottom of Page 12), as long as the noise matrix $E$ does not change the sub-orders of $C^*$, $\operatorname{sign}(L_f C^*)$ will be the same as $\operatorname{sign}(L_f (C^*+E))$ and thus the loss would not be affected. We'll add more details for the proof of Lemma 2 and Lemma 3 in the revised version to make them clearer and more rigorous.
> >
> > ---
> > ### References
> >
> > [1] Ghosh, Dibya, Marlos C Machado, and Nicolas Le Roux. "An operator view of policy gradient methods." _Advances in Neural Information Processing Systems_ 33 (2020).
> >
> > [2] Abdolmaleki, Abbas, Jost Tobias Springenberg, Yuval Tassa, Remi Munos, Nicolas Heess, and Martin Riedmiller. "Maximum a posteriori policy optimisation."
> > _International Conference on Learning Representations (ICLR)_ (2018).
> >
> > [3] Falkner, Stefan, Aaron Klein, and Frank Hutter. "BOHB: Robust and efficient hyperparameter optimization at scale." _International Conference on Machine Learning (ICML)_, pp. 1437-1446. (2018).

---

> > > ### Comment · Reviewer_8c8S · 2021-11-21
> > > **Reply to the authors**
> > >
> > > Thank you for the detailed response.

---

### Decision · Program_Chairs · 2022-01-20

**Decision:**

Accept (Poster)

**Comment:**

This paper studies the problem called "Predict, then Optimize (PTO)", where the goal is to predict some unknown parameters in the objective function and then optimize the function. The paper focus on special case of PTO where the goal is to learn the unknown parameters corresponding to some combinatorial ordering of the data. The paper argues that traditional methods which perform the ranking after minimizing the objective can be very unstable, and proposes a new alternative loss called ATGP loss. The goal of the loss is to ensure that the total group preorder of the learned solution is as consistent with the observation as possible after fitting the data.

The paper shows some property of the new loss and claims that it can be better than naive regression then ranking approach on learning to rank with linear regression problem, however, the theorem only shows that the robustness bar of TGP loss is larger or EQUAL to that of the naive approach, not exactly higher -- Thus, such an argument is not really convincing.  The authors also proposed a heuristic algorithm to minimize a smoothed version of the proposed loss.

However, the proposed objective is still interesting on its own, and the experimental result looks promising. Thus, this paper is a borderline accept paper at this conference.